# Comparison of Various Solvent Extracts and Major Bioactive Components from Unsalt-Fried and Salt-Fried Rhizomes of *Anemarrhena asphodeloides* for Antioxidant, Anti-α-Glucosidase, and Anti-Acetylcholinesterase Activities

**DOI:** 10.3390/antiox11020385

**Published:** 2022-02-14

**Authors:** Yi-Cheng Chu, Chang-Syun Yang, Ming-Jen Cheng, Shu-Ling Fu, Jih-Jung Chen

**Affiliations:** 1Institute of Traditional Medicine, School of Medicine, National Yang Ming Chiao Tung University, Taipei 112, Taiwan; chuyc.md07@nycu.edu.tw; 2Department of Pharmacy, School of Pharmaceutical Sciences, National Yang Ming Chiao Tung University, Taipei 112, Taiwan; tim0619@nycu.edu.tw; 3Bioresource Collection and Research Center (BCRC), Food Industry Research and Development Institute (FIRDI), Hsinchu 300, Taiwan; cmj@firdi.org.tw; 4Department of Medical Research, China Medical University Hospital, China Medical University, Taichung 404, Taiwan

**Keywords:** *Anemarrhena asphodeloides*, various solvent extracts, antioxidant activity, anti-α-glucosidase activity, anti-acetylcholinesterase activity

## Abstract

The rhizome of *Anemarrhena asphodeloides* Bunge (AA, family Liliaceae) is a famous and frequently used herbal drug in the traditional medicine of Northeast Asia, under vernacular name “zhimu”. *A. asphodeloides* has been used as an anti-inflammatory, antipyretic, anti-platelet aggregation, anti-depressant, and anti-diabetic agent in traditional Chinese medicine. We examined the antioxidant, anti-acetylcholinesterase (AChE), and anti-α-glucosidase activities of various solvent extracts and the main bioactive compounds from the rhizome of *A. asphodeloides*. Acetone extract exhibited comparatively high antioxidant activities by 2,2-diphenyl-1-(2,4,6-trinitrophenyl)hydrazyl (DPPH), 2,2′-azino-bis(3-ethylbenzothiazoline-6-sulfonic acid) (ABTS) radical scavenging, and ferric-reducing antioxidant power (FRAP) assays. A water extract exhibited relatively strong antioxidant activity by superoxide radical scavenging test. Furthermore, dichloromethane, chloroform, and *n*-hexane extracts showed significant anti-α-glucosidase activities. Finally, ethanol and dichloromethane extracts exhibited relatively strong AChE inhibitory activity. HPLC analysis was used to examine and compare various solvent extracts for their compositions of isolates. We isolated four major chemical constituents and analyzed their antioxidant, anti-α-glucosidase, and AChE inhibitory activities. The bioactivity assays showed that mangiferin displayed the most potential antioxidant activities via FRAP, ABTS, DPPH, and superoxide assays and also exhibited the most effective anti-AChE and anti-α-glucosidase activities among all the isolates. The present study suggests that *A. asphodeloides* and its active extracts and components are worth further investigation and might be expected to develop as a candidate for the treatment or prevention of oxidative stress-related diseases, AChE inhibition, and hyperglycemia.

## 1. Introduction

Diabetic treatment is based on carbohydrate enzymes (such as α-glucosidase and α-amylase) inhibition as well as other mechanisms, such as reduction of glucose level and maintenance of β-cell performance. It has been shown that the inhibition of α-glucosidase and α-amylase correlates with an increase in postprandial glucose level and reduces the absorbance of glucose in the intestine, limiting the excursion of glucose in plasma. The control of activity is an important feature in diabetes treatment [1].

Oxidative stress and reactive oxygen species (ROS) play an important role in some diseases, such as Alzheimer’s disease (AD) and diabetes. Oxidative stress creates an excess of ROS/nitrogen species, leading to injury of the cell composition (such as lipids, protein enzymes, and nucleic acids) that may have a predictable aftermath in neurodegenerative disorders such as Alzheimer’s [2] and metabolic ailments such as diabetes [3]. The use of acetylcholinesterase inhibitory drugs is the most appropriate modern and primary treatment method to confront neurodegenerative diseases [4]. Because the major enzyme in the pathogenesis of Alzheimer’s disease is AChE, the suppression of AChE raises the signal transfer in cholinergic pathways and decreases the symptoms of Alzheimer’s disease [5]. The use of synthetic drugs in therapy for AD such as galantamine has adverse reactions, including weight loss, diarrhea, loss of appetite, nausea, vomiting, dizziness, headache, stomach pain, and muscle weakness [6].

However, treatment of diabetes and AD with drugs such as galantamine is no longer reassuring and without side effects. In fact, many synthetic drugs may induce several undesirable symptoms, like gastrointestinal troubles or abdominal disorders [7]. Assuredly, natural products are favorable drug candidates because they are easy to get and relatively safe [8].

*Anemarrhena asphodeloides*, known as “zhimu” in Taiwan, which is listed in the Taiwan herbal pharmacopeia, is used to clear heat and resolve toxins. *A. asphodeloides* has also been found to have antifungal activity [9], decrease the blood glucose concentration [10], inhibit platelet agglutination and neoplastic activity [11,12], act as an antiallergic drug [13], and be an anti-dementia agent [14]. Furthermore, it has been reported that the pharmacological effects of *A. asphodeloides* are tightly connected to the methods of medicinal material processing. In other words, different processing methods will get different pharmacological effects [15]. In this study, the antioxidant, anti-α-glucosidase, and anti-AChE activities of *A. asphodeloides* (Figure 1) and its active extracts and components were examined.

## 2. Materials and Methods

### 2.1. Chemicals and Reagents

Folin–Ciocalteau reagent, 2,2′-azino-bis(3-ethylbenzothiazoline-6-sulfonic acid) (ABTS), ascorbic acid, ethylenediaminetetraacetic acid (EDTA), α-glucosidase, acetylcholinesterase (AChE), acetylcholine iodide (AchI), 5,5′-dithiobis(2-nitrobenzoic acid) (DTNB), 2,4,6-Tris(2-pyridyl)-s-triazine (TPTZ), and Trolox were purchased from Sigma-Aldrich (St. Louis, MO, USA). Galanthamine and quercetin were purchased from MedChemExpress (Monmouth Junction, NJ, USA). *p*-Nitro-phenyl-α-D-glucopyranoside (*p*-NPG), aluminum chloride (AlCl_3_), and iron (III) chloride were purchased from Alfa Aesar (Lancashire, UK). Sodium carbonate, potassium peroxodisulfate, potassium dihydrogenphosphate, and disodium hydrogenphosphate were supplied by SHOWA Chemical Co., Ltd. (Chuo-ku, Japan). Loroglucinol, 2,2-diphenyl-1-(2,4,6-trinitrophenyl) hydrazyl (DPPH), phenazine methosulphate (PMS), nitroblue tetrazolium (NBT), and deoxyribose were purchased from Tokyo Chemical Industry Co., Ltd. (Tokyo, Japan). Butyl hydroxytoluene (BHT), nicotinamide adenine dinucleotide (NADH), sodium acetate, and potassium acetate were obtained from Acros Organics (Geel, Belgium). Acetic acid was purchased from Macron Fine Chemicals (Center Valley, PA, USA).

### 2.2. Preparation of A. asphodeloides Extract

The rhizomes of *A. asphodeloides* were purchased from Wanhua Dist., Taipei City, Taiwan, in July 2021 and identified by Prof. J.-J. Chen. A voucher specimen was deposited in the Department of Pharmacy, National Yang Ming Chiao Tung University, Taipei, Taiwan. Preparation of *A. asphodeloides* extract was performed as described previously [16].

### 2.3. Preparation of Active Components

The rhizomes (1.2 kg) of *Anemarrhena asphodeloides* were extracted and shredded with MeOH (3 × 1.2 L, 3 d each) at room temperature. The MeOH extract was concentrated under reduced pressure at 37 °C, and the MeOH extract (420.81 g) was obtained. Part (105 g) of the MeOH extract was purified by reversed-phase column chromatography (CC) (4.7 kg of reversed-phase silica gel, 200–400 mesh; water/methanol gradient) to afford 12 fractions: A1–A12. Fraction A3 (5.35 g) was purified by reversed-phase column chromatography (CC) (240 g of reversed-phase silica gel, 70–230 mesh; water/acetonitrile gradient) to afford 12 fractions: A3-1–A3-12. Part (138 mg) of fraction A3-6 was further purified by preparative TLC (RP-18; water/acetonitrile, 1:2) to afford neomangiferin (2.04 mg) (R_f_ = 0.68), isomangiferin (3.24 mg) (R_f_ = 0.54), and timosaponin AIII (3.98 mg) (R_f_ = 0.32). Part (115 mg) of fraction A3-9 was further purified by preparative TLC (RP-18; water/acetonitrile, 3:2) to obtain mangiferin (4.17 mg) (R_f_ = 0.42). The structures of neomangiferin, isomangiferin, timosaponin AIII, and mangiferin were identified by nuclear magnetic resonance (NMR) spectra acquired using a Bruker Avance 400 MHz spectrometer (Bruker, Bremen, Germany).

### 2.4. Reverse-Phase HPLC

The assay for measuring reverse-phase HPLC of four components was carried out as described previously with a slight modification [16]. In order to obtain a suitable chromatographic condition, the mobile phase conditions were optimized. The mobile phase consisting of 0.2% acetic acid in water and acetonitrile showed better separating selectivity than that consisting of methanol and water. Gradient separation using 0.2% acetic acid in water (*v*/*v*) (solvent A) and acetonitrile (solvent B) as mobile phase was as follows: 0–3 min, 100% A with isocratic elution; 3–9 min, linear gradient from 0 to 5% B; 9–22 min, linear gradient from 5 to 13% B; 22–32 min, linear gradient from 13 to 16% B; 32–45 min, linear gradient from 16 to 35% B; 45–60 min, linear gradient from 35 to 65% B; 60–80 min, linear gradient from 65 to 80% B; 80–85 min, linear gradient from 80 to 100% B; 85–95 min, back to initial conditions at 0% B; and 95–105 min, at 0% B. The flow rate was 1.0 mL/min and the injection volume was 10 μL. Peaks were detected at 280 nm. Quantification of four components from *Anemarrhena asphodeloides* in each solvent extract was performed as described above.

### 2.5. Determination of Total Phenolic Content

Total phenolic content (TPC) of various solvent extracts and major bioactive components was performed according to the method previously reported [16].

### 2.6. Determination of Total Flavonoid Content

The total flavonoid content (TFC) was conducted by the reference method with a slight modification [17]. In short, the extracted sample was diluted with MeOH to reach a concentration of 100 μg/mL. Quercetin was diluted in MeOH (0–100 μg/mL) as standard. The diluted extract sample or quercetin (400 µL) was mixed with 10% (*w*/*v*) aluminum chloride solution (200 µL) and 0.1 mM potassium acetate solution (200 µL). The mixture was reacted at room temperature for 30 min. Then the absorbance of the mixture was measured at 415 nm. The TFC of the extracts was determined from a standard calibration curve using quercetin. TFC was expressed as milligram quercetin equivalent per gram *A. asphodeloides* (mg QCE/g AA).

### 2.7. DPPH Radical Scavenging Activity

The DPPH radical scavenging assay was measured by the reference method [18].

### 2.8. ABTS Radical Scavenging Activity

ABTS radical scavenging activity of each extract was determined as previously described [19].

### 2.9. Superoxide Radical Scavenging Activity

Superoxide anion radical (O_2_^•–^) scavenging activity was measured using the previously described method [20].

### 2.10. Ferric-Reducing Antioxidant Power (FRAP) Assay

The FRAP assay was determined by the reference method with a slight modification [21,22]. The working solution was mixed with acetate buffer (pH 3.6), ferric chloride solution (20 mM), and TPTZ solution (10 mM TPTZ in 40 mM HCl) in a proportion of 10:1:1, respectively, and freshly prepared before being used. A total of 900 μL of the working solution was warmed to 37 °C and then mixed with 100 μL of the diluted sample, blank or standard, in a microcentrifuge tube. The tubes were vortexed in a dry bath at 37 °C for 40 min. The absorbance was measured at 593 nm. The standard curve was linear between 0 and 100 mM Trolox. Results are expressed in mM TE/g dry weight. Additional dilution was needed if the FRAP value measured was over the linear range of the standard curve.

### 2.11. α-Glucosidase Inhibitory Activity Assay

The inhibition assay of α-glucosidase was conducted using the procedure previously reported [23].

### 2.12. Acetylcholinesterase Inhibitory Activity Assay

The inhibition assay of acetylcholinesterase was conducted using the conditions previously reported with slight modifications [24]. Briefly, 0.1 M sodium phosphate buffer (pH 8.0, 140 μL), test compound solution (20 μL), enzyme solution (15 μL of AChE 0.2 units/mL solution), and DTNB (15 mM, 10 μL) were mixed and incubated for 10 min at room temperature. Substrate (10 μL of AchI 15 mM solution) was then added and the reaction was initiated.

### 2.13. Molecular Modeling Docking Study

The in silico evaluation was conducted in AutoDock Vina software [25]. The crystal structure (PDB: 3A4A) was retrieved from the Protein Databank and hydrogen atoms were added to prepare the docked receptor. The 3D structures of ligands were constructed in the Chem3D program. The hydrogen supplement, Gasteiger charge measurement for protein atoms, and selection of flexible torsions for ligands were conducted by AutodockTools (ADT ver. 1.5.6). The size of the grid was designed at 20 Å × 26 Å × 22 Å of mangiferin, 30 Å × 30 Å × 30 Å of timosaponin A-III, 20 Å × 25 Å × 20 Å of neomangiferin, 18 Å × 18 Å × 18 Å of isomangiferin, and 20 Å × 20 Å × 20 Å of acarbose. Additionally, the grid center at dimensions (x, y, and z, respectively) 21.2, −0.8, and 18.6 was determined. The binding affinity energy was provided as docking scores and is shown in kcal/mol. The best interaction was considered only the top-scoring pose. The visualization of the best docking interactions was analyzed in Biovia Discovery Studio client 2021 [26].

### 2.14. Statistical Analysis

All data are expressed as mean ± SEM. Statistical analysis was carried out using Student’s *t*-test. A probability of 0.05 or less was considered statistically significant. All the experiments were performed at least 3 times.

## 3. Results

### 3.1. Determination of Total Phenolic Content (TPC), Total Flavonoid Content (TFC), and Yields in Each Solvent Extract

We studied the TPC, TFC, and yields in various solvent extracts of *A. asphodeloides* (AA) and salt-fried *Anemarrhena asphodeloides* (salt-fried AA). Table 1 displays TPC, TFC, and extraction yields of *n*-hexane, chloroform, dichloromethane, ethyl acetate, acetone, methanol, ethanol, and water extracts from *A. asphodeloides*. The yields of different solvent extracts ranged from 0.3 ± 0.1% (*n*-hexane extract) to 76.7 ± 4.4% (water extract) of AA, and 0.3 ± 0.1% (*n*-hexane extract) to 72.9 ± 2.5% (water extract) of salt-fried AA. The water extract exhibited the largest yield among all extracts possibly due to its abundant amounts of high polar compounds. Significant differences were found in TPC among all various solvent extracts, of which acetone extract from AA contained the largest amount of TPC (45.3 ± 1.2 mg/g). In addition, ethyl acetate extract from salt-fried AA contained the highest amount of TPC (78.2 ± 3.0 mg/g). The TFC among different solvent extracts ranged from 5.8 ± 2.2% (ethyl acetate extract) to 42.9 ± 4.9% (*n*-hexane extract) of AA, and 8.3 ± 1.4% (*n*-hexane extract) to 45.4 ± 4.2% (acetone extract) of salt-fried AA.

### 3.2. DPPH Free-Radical Scavenging Activity

The DPPH radical scavenging activity of each extract is shown in Table 2, and butylated hydroxytoluene (BHT) was used as the positive control. From the results of our tests, acetone extract (IC_50_ = 123.6 ± 3.9 μg/mL) of AA and water extract (IC_50_ = 113.9 ± 8.6 μg/mL) of salt-fried AA exhibited relatively strong antioxidant activities by DPPH radical scavenging assay.

### 3.3. ABTS Free-Radical Scavenging Activity

As shown in Table 2, among the different solvent extracts of *A. asphodeloides*, acetone (IC_50_ = 48.5 ± 2.9 μg/mL) exhibited the highest ABTS radical scavenging activity followed by chloroform (IC_50_ = 73.8 ± 2.3 μg/mL), ethyl acetate (IC_50_ = 75.3 ± 3.7 μg/mL), dichloromethane (IC_50_ = 95.4 ± 5.5 μg/mL), water (IC_50_ = 117.9 ± 0.6 μg/mL), ethanol (IC_50_ = 177.8 ± 3.4 μg/mL), methanol (IC_50_ = 216.0 ± 3.9 μg/mL), and *n*-hexane (IC_50_ = 305.8 ± 10.4 μg/mL). Furthermore, among the different solvent extracts of salt-fried *A. asphodeloides*, ethyl acetate (IC_50_ = 24.1 ± 1.2 μg/mL) exhibited the highest ABTS radical scavenging activity, followed by dichloromethane (IC_50_ = 40.9 ± 2.5 μg/mL), acetone (IC_50_ = 41.9 ± 0.7 μg/mL), chloroform (IC_50_ = 64.3 ± 3.9 μg/mL), water (IC_50_ = 97.5 ± 1.6 μg/mL), ethanol (IC_50_ = 112.4 ± 3.6 μg/mL), and methanol (IC_50_ = 119.4 ± 4.2 μg/mL).

### 3.4. Superoxide Radical Scavenging Activity

Obviously, the result displaying all extracts had no significant effect on superoxide radical scavenging activity (IC_50_ > 400 μg/mL), except for the water extract from AA (IC_50_ = 369.9 ± 4.3 μg/mL) and the water extract from salt-fried AA (IC_50_ = 296.4 ± 6.1 μg/mL) (Table 2).

### 3.5. Ferric-Reducing Antioxidant Power (FRAP) Assay

The FRAP assay of each extract is shown in Table 2, and butylated hydroxytoluene (BHT) was used as the positive control. The FRAP assay is expressed as millimolar (mM) of Trolox equivalents (TE) per gram of extract. From the results of our tests, acetone extract (381.5 ± 8.3 mM TE/g) of AA and ester acetate extract (485.5 ± 4.3 mM TE/g) of salt-fried AA exhibited relatively high antioxidant powers via FRAP assays.

Based on the above data of ABTS, DPPH radical scavenging activity, and FRAP assays, acetone extract displayed the strongest antioxidant activity of all the solvent extracts. Additionally, water extract exhibited relatively strong antioxidant activity by superoxide radical scavenging test.

### 3.6. Anti-α-Glucosidase Activity Assay

As shown in Table 3, the dichloromethane extract of *A. asphodeloides* exhibited the most anti-α-glucosidase activity (IC_50_ = 21.8 ± 1.4 μg/mL), followed by chloroform (IC_50_ = 23.4 ± 2.3 μg/mL), *n*-hexane (IC_50_ = 25.9 ± 2.0 μg/mL), ethyl acetate (IC_50_ = 55.5 ± 2.8 μg/mL), and acetone (IC_50_ = 101.7 ± 15.9 μg/mL). Furthermore, among the different extracts of salt-fried *A. asphodeloides*, ethyl acetate (IC_50_ = 26.0 ± 0.6 μg/mL) exhibited the highest anti-α-glucosidase activity, followed by dichloromethane (IC_50_ = 32.0 ± 2.5 μg/mL), chloroform (IC_50_ = 34.7 ± 4.7 μg/mL), *n*-hexane (IC_50_ = 34.8 ± 3.0 μg/mL), and acetone (IC_50_ = 105.0 ± 2.6 μg/mL).

The solvents extracted by *n*-hexane, chloroform, dichloromethane, ethyl acetate, and acetone were more effective than the positive control, acarbose (IC_50_ = 305.0 ± 4.6 μg/mL). Among all solvent extracts, dichloromethane extract of AA and ethyl acetate extract of salt-fried AA showed the highest anti-α-glucosidase activity. These results show that lower polar solvent extracts of *A. asphodeloides* possessed a higher α-glucosidase inhibitory effect.

### 3.7. Acetylcholinesterase (AChE) Inhibitory Activity Assay

The AChE inhibitory activity of each extract is shown in Table 4, and galanthamine was used as the positive control. From the results of our tests, ethanol extract (IC_50_ = 117.1 ± 2.6 μg/mL) of AA and water extract (IC_50_ = 73.6 ± 2.8 μg/mL) of salt-fried AA exhibited relatively high AChE inhibitory activities among all solvent extracts.

### 3.8. Quantitation of Active Components in Different Solvent Extracts

Appendix A display the quantification of active components in different solvent extracts from *Anemarrhena asphodeloides* by reverse-phase HPLC analyses. The contents of four active components in each solvent extract are shown in Table 5. The total quantity of four bioactive components in each extract of AA ranged from a maximum of 47.5 ± 1.5 mg/kg (chloroform extract) (Figure 2) to a minimum of 9.6 ± 0.5 mg/kg (ethanol extract) in the following order: chloroform > *n*-hexane > acetone > dichloromethane > water > ethyl acetate > methanol > ethanol extract. Additionally, the total quantity of four bioactive components in each extract of salt-fried AA ranged from a maximum of 46.5 ± 1.4 mg/kg (chloroform extract) (Figure 3) to a minimum of 12.6 ± 0.8 mg/kg (water extract) in the following order: chloroform > *n*-hexane > ethyl acetate > dichloromethane > acetone > methanol > ethanol > water extract. Chloroform (47.5 ± 1.5 mg/kg in AA and 46.5 ± 1.4 mg/kg in salt-fried AA) extracts exhibited larger amounts of four active components compared with other extracts. Isomangiferin was the most abundant among the four active compounds in organic solvent extract, followed by timosaponin A-III, neomangiferin, and mangiferin.

### 3.9. Antioxidant Activities of Isolated Components

The isolated compounds mangiferin, timosaponin A-III, neomangiferin, and isomangiferin (Figure 4) were measured for their antioxidant activities, including DPPH, ABTS, superoxide radical scavenging activity, and FRAP assays. Results are shown in Table 6. Mangiferin (IC_50_ = 5.4 ± 0.3 μg/mL) exhibited the highest DPPH radical scavenging activity, followed by isomangiferin (IC_50_ = 16.7 ± 1.1 μg/mL), timosaponin A-III (IC_50_ > 200 μg/mL), and neomangiferin (IC_50_ > 200 μg/mL). Mangiferin (IC_50_ = 3.7 ± 0.2 μg/mL) also exhibited stronger antioxidant activity than other isolates by ABTS and superoxide radical scavenging assays. Furthermore, mangiferin (9371.4 ± 183.9 mM TE/g) exhibited relatively higher antioxidant activity than others via FRAP assay.

According to the above results, the acetone extract of *A. asphodeloides* and ethyl acetate extract of salt-fried *A. asphodeloides* contained the highest amount of main phenolic components and therefore the highest antioxidant power.

### 3.10. Anti-α-Glucosidase Activities of Isolated Components

For further discussion of the α-glucosidase inhibitory activity, we conducted further investigation on the main components isolated from *A. asphodeloides*. As shown in Table 7, mangiferin exhibited the strongest anti-α-glucosidase activity (IC_50_ = 61.6 ± 5.1 μg/mL), followed by timosaponin A-III (IC_50_ = 72.5 ± 1.9 μg/mL), isomangiferin (IC_50_ = 183.2 ± 1.3 μg/mL), and neomangiferin (IC_50_ > 400 μg/mL). Additionally, the major components, mangiferin, timosaponin A-III, and isomangiferin, displayed more effective anti-α-glucosidase activities than the positive control, acarbose (IC_50_, 322.0 ± 18.7 μg/mL).

### 3.11. Acetylcholinesterase (AChE) Inhibitory Assays of Isolated Components

For further evaluation of the AChE inhibitory activity, we conduct anti-AChE assays of the main components isolated from *A. asphodeloides*. Among the isolated compounds, mangiferin exhibited higher anti-AChE activity than other isolates. Table 8 shows the AChE inhibitory effects of the major active components from *A. asphodeloides*.

### 3.12. Molecular Docking Study

The 3D crystal structure of α-glucosidase showed that it mainly contained numerous structural domains, including the N-terminal domain, the barrel domain where the active site is located, and the C-terminal domain. The α-glucosidase active site was primarily formed by numerous β-sheets and several loops or α-helices. Importantly, the active site in different species showed certain conformational similarities. It mainly contained hydrophilic residues and allowed compounds with distinct sizes to enter. To further study how compounds might interact with α-glucosidase of *Saccharomyces cerevisiae* to exhibit its antagonistic effect, the docking models of compounds were generated with the Discovery Studio 2021 (Accelrys, San Diego, CA, USA) modeling program. According to the anti-α-glucosidase experimental data, mangiferin (Figure 5), timosaponin A-III (Figure 6), isomangiferin (Figure 7), neomangiferin (Figure 8), and acarbose (Figure 9) were selected to determine their binding abilities to the crystal structure of isomaltase from *Saccharomyces cerevisiae*.

The 3D crystal structure for α-glucosidase of *Saccharomyces cerevisiae* is not available at this moment, so the crystal structure (PDB: 3A4A) of *Saccharomyces cerevisiae* (PDB: 3A4A) containing 72% sequence homology with α-glucosidase from *Saccharomyces cerevisiae* is usually used to perform the docking study and was also employed in this study. In the crystal structure (PDB: 3A4A), the configuration of its active site is quite similar to that of α-glucosidase from beta vulgaris, but it is deep and narrow. The crystal structure of this active site reveals that its co-crystallized ligand, α-D-glucopyranose, is located deep in the ligand-binding pocket and makes three essential H-bond interactions, including (1) the 4-hydroxyl group interacting with His 351 and Asp 352 by acting as the H-bond donor; (2) the 5-hydroxyl group making H-bond contact Glu 277 as well as Asp 352 by acting as the H-bond donor, and also serving as the H-bond acceptor to interact with Arg 213; and (3) the 6-hydroxymethyl group acting as the H-bond donor to interact with Glu 277 and Asp 352.

Before the docking simulation, the ligand (α-D-glucopyranose) included in the 3A4A PDB file was redocked for validation. The interactions between alpha-D-glucopyranose and 3A4A and the best pose of the calculated results showed high similarity and repeatability with native data (data not shown). The results indicate the high accuracy of the existing simulation system and supported further computing.

The lowest binding energy of each ligand was considered the best conformation. The binding affinities are listed in Table 9. In this study, acarbose was used as positive control. Compared with acarbose, the binding energies of mangiferin, timosaponin A-III, and isomangiferin were lower than −2.3 kcal/mol (Table 9). This suggests that mangiferin, timosaponin A-III, and isomangiferin could dock into the pocket of the crystal structure of isomaltase from *Saccharomyces cerevisiae* similar to or even better than that of acarbose.

As show in Figure 5, mangiferin was bound with Ser 311, Asp 307, Gln 279, and Asp 215 through conventional hydrogen bonds, and other interactions (π-anion, π-π T-shaped, and π-alkyl) were also observed with Asp 352, Phe 178, Tyr 158, and Val 216. Arg 315 formed a carbon hydrogen bond of mangiferin. In addition, the two hydroxyl groups provided unfavorable donor–donor interactions with Arg 442 and His 280 residues. These allowed mangiferin and protein to form a stable complex.

Timosaponin A-III also established carbon hydrogen bonds with Asp 352, Pro 312, and Ser 304, together with other interactions (alkyl and π-alkyl) with Tyr 158, Val 216, Phe 178, Phe 303, Arg 315, and His 280 residues of the *S. cerevisiae* α-glucosidase being detected. Additionally, unfavorable acceptor–acceptor and unfavorable donor–donor interactions were found with Thr 310 and Gly 309 (Figure 6).

Finally, isomangiferin was bound with Asp 242 and Arg 442 through conventional hydrogen bonds, and other interactions (π-anion and π-π stacked) were also observed with Glu 277 and Phe 303. In addition, unfavorable acceptor–acceptor interactions were observed with Glu 353 (Figure 7).

According to our data, the docking scores of mangiferin, timosaponin A-III, and isomangiferin were higher than those of neomangiferin and acarbose, which suggests better binding capability. In this study, the active ingredients mangiferin, timosaponin A-III, and isomangiferin possessed not only anti-α-glucosidase activity, but also better binding potential with the active sites of *S. cerevisiae* α-glucosidase. This indicates that these compounds may deserve further investigation as natural α-glucosidase inhibitors.

## 4. Discussion

Diverse methods have been utilized and developed for the extraction of natural products (i.e., herbs, plants, and fungi) for application as alternatives to modern medicines. The most common method that has been applied for many years is boiling or making a decoction with water, a very easy and economically feasible process. Nowadays, medicinal plant-based studies have been conducted and many are ongoing, as various metabolites with health benefits have been found in natural products. Recently, in more advanced studies, organic solvents are used to get natural product extracts, including a variety of metabolites, according to the polarity and property of the component of interest [27]. Other factors that may affect the process applied to natural product extraction comprise the kind of solvent to be used, the temperature set during extraction, the properties of the plant material, and the target metabolites [28]. The change in solvent polarity results in a significant difference in phytochemical components and bioactivities. Thus, we utilized solvents of various polarity to get the rhizome fractions from *A. asphodeloides* in an effort to assess these different metabolites. We found various metabolites with different degrees of bioactivity due to their differences in solvent polarity.

ABTS and DPPH assays have been widely utilized to assess the antioxidant activities of natural components. Both assays are mostly connected with the proton radical scavenging or hydrogen donating capacities of the target compounds [29]. Superoxide radical scavenging activity is measured by the PMS/NADH–NBT system. Superoxide anion radicals generated from dissolved oxygen by PMS/NADH coupling reaction reduce NBT. The decrease in absorbance at 560 nm with antioxidants indicates a reduction in superoxide anion radicals in the reaction mixture [30]. The ferric-reducing antioxidant power (FRAP) measures the antioxidant potential of each extract through the reduction of ferric iron (Fe^3+^) complex to ferrous iron (Fe^2+^) complex by antioxidants present in the samples [31]. In our study, ethyl acetate and acetone extract in salt-fried rhizomes of *A. asphodeloides* displayed high antioxidant activities among all solvent extracts via ABTS and FRAP assays, which may be connected to TPC in the extracts, and the ethyl acetate extract showed the strongest antioxidant activity. The differences in antioxidant capacities of the extracts may be due to the different extents of TPC or the composition of antioxidant compounds in the extracts.

The comparative evaluation of total phenolic content (TPC), total flavonoid content (TFC), and antioxidant assays (DPPH, ABTS, superoxide, and FRAP) of various solvent extracts (*n*-hexane, chloroform, dichloromethane, EtOAc, acetone, EtOH, MeOH, and water) from the salt-fried and non-salt-fried rhizomes of *A. asphodeloides* was first mentioned in this study. This can provide a guide for the selection of appropriate solvents in TPC, TFC, and antioxidant extraction applications. Based on the antioxidant data, mangiferin displayed potent antioxidant properties. Similar experimental results could also be found in past studies [32]. Antioxidant assays of isomangiferin (FRAP), neomangiferin (superoxide), and timosaponin A-III (DPPH, ABTS, superoxide, and FRAP) were first evaluated in this study.

An anti-α-glucosidase agent could suppress the activity of glucosidases in the small intestine, which split the glycosidic bonds in carbohydrates so that it decreases the glucose release from food. The inhibitors studied were classified into non-sugar and sugar-mimicking types on the basis of their chemical structure. The anti-α-glucosidase drugs for clinical treatment such as voglibose, acarbose, and miglitol all belong to the sugar-mimicking type. Nevertheless, the α-glucosidase inhibitors with the non-sugar type have received the attention of researchers due to the limitations of sugar-mimicking inhibitors. Mangiferin possesses a similar basic xanthone backbone, which is the molecular basis for anti-α-glucosidase [33]. In anti-α-glucosidase assay, mangiferin, timosaponin A-III, and isomangiferin showed more effective anti-α-glucosidase activities than acarbose (positive control). Our data show that mangiferin was about five-fold stronger than acarbose against α-glucosidase.

According to the results of the anti-α-glucosidase assay, mangiferin exhibited the most potent anti-α-glucosidase activity among all isolated compounds. Thus, the interaction between α-glucosidase and mangiferin was evaluated by molecular modeling docking. As a result of molecular docking, mangiferin exhibited high affinity with α-glucosidase. Similar experimental results could also be found in past studies [34,35]. Molecular docking study for timosaponin A-III and isomangiferin with *S. cerevisiae* α-glucosidase was first evaluated in this study. Furthermore, the binding energies of mangiferin, timosaponin A-III, neomangiferin, and isomangiferin with *S. cerevisiae* α-glucosidase were also first calculated in our study.

Alzheimer’s disease (AD) is a progressive degenerative disease of the brain that is characterized by the deterioration of cognitive and memory functions. Treatments have focused on increasing brain cholinergic activity via AChE inhibitors. Actually, an AChE inhibitor, galanthamine, has been approved to treat AD [14,36]. In previous studies, mangiferin was supposed to possess some potential in postponing the progression of memory loss in AD. Mangiferin has drawn much attention from many researchers, as it may be able to improve memory function of the brain via diverse mechanisms [37]. In our study, mangiferin and neomangiferin also exhibited anti-AChE activity and deserve further study.

## 5. Conclusions

Various solvent extracts of *A. asphodeloides* were investigated with various antioxidant systems, anti-α-glucosidase, and acetylcholinesterase (AChE) inhibitory activity assays. In our study, ethyl acetate extract of salt-fried *A. asphodeloides* displayed the strongest antioxidant activity among all solvent extracts via ABTS and FRAP assays. Dichloromethane extract of *A. asphodeloides* showed the highest anti-α-glucosidase activity among all solvent extracts. Water extract of salt-fried *A. asphodeloides* exhibited the most potential DPPH and superoxide radical scavenging and anti-AChE activities among all solvent extracts. Four isolated compounds from *A. asphodeloides* were quantified by HPLC and identified as mangiferin, timosaponin A-III, neomangiferin, and isomangiferin. Moreover, the comparative evaluation for the identification and quantification of the major active components (mangiferin, timosaponin A-III, neomangiferin, and isomangiferin) of different solvent extracts (*n*-hexane, chloroform, dichloromethane, EtOAc, acetone, EtOH, MeOH, and water) from the salt-fried and non-salt-fried rhizomes of *A. asphodeloides* by HPLC analysis was first conducted in our study.

The bioactivity assays showed that mangiferin displayed the most potential antioxidant activity via DPPH, ABTS, superoxide, and FRAP assays and also exhibited the most effective anti-α-glucosidase and anti-AChE activities among all isolates. Mangiferin, timosaponin A-III, and isomangiferin possess significant anti-α-glucosidase potential. Further molecular docking computing results supported mangiferin, timosaponin A-III, and isomangiferin exhibiting a binding affinity to the active pocket of *S. cerevisiae* α-glucosidase.

The above active extracts and their active components can be used as potential natural antioxidants. Furthermore, mangiferin also can be used as a natural anti-α-glucosidase and anti-AChE agent.

## Figures and Tables

**Figure 1 antioxidants-11-00385-f001:**
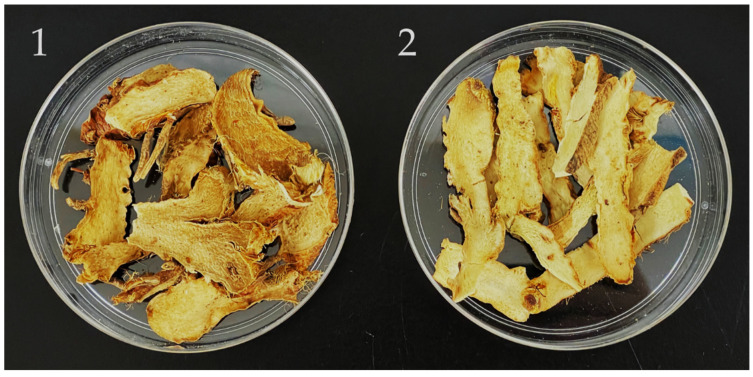
Non-salt-fried (1) and salt-fried (2) rhizomes of *Anemarrhena asphodeloides* were used in the study.

**Figure 2 antioxidants-11-00385-f002:**
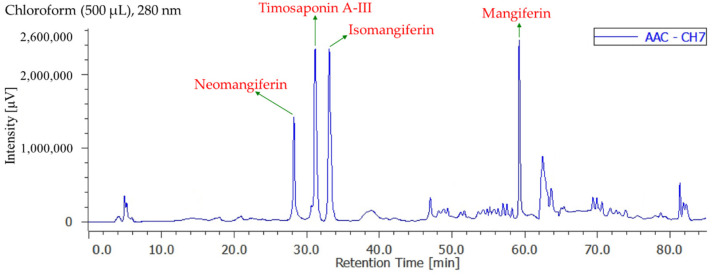
Reverse-phase HPLC chromatogram of chloroform extract in non-salt-fried rhizomes of *A. asphodeloides*.

**Figure 3 antioxidants-11-00385-f003:**
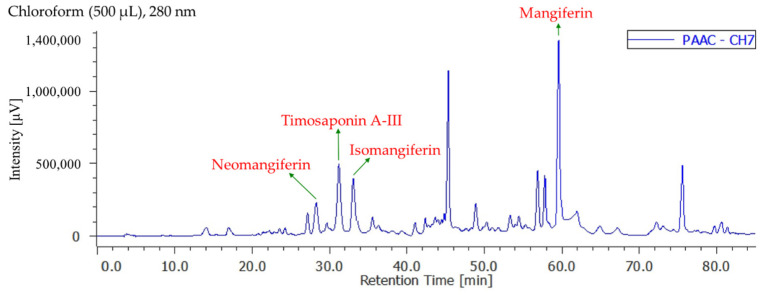
Reverse-phase HPLC chromatogram of chloroform extract in salt-fried rhizomes of *A. asphodeloides*.

**Figure 4 antioxidants-11-00385-f004:**
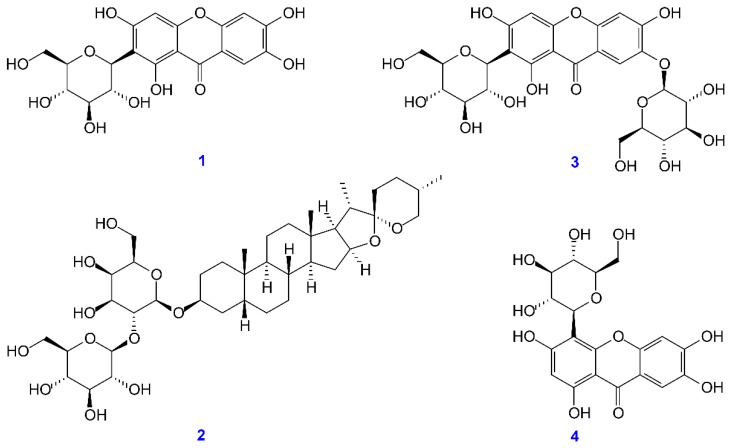
Chemical structures of mangiferin (**1**), timosaponin A-III (**2**), neomangiferin (**3**), and isomangiferin (**4**) from *Anemarrhena asphodeloides*.

**Figure 5 antioxidants-11-00385-f005:**
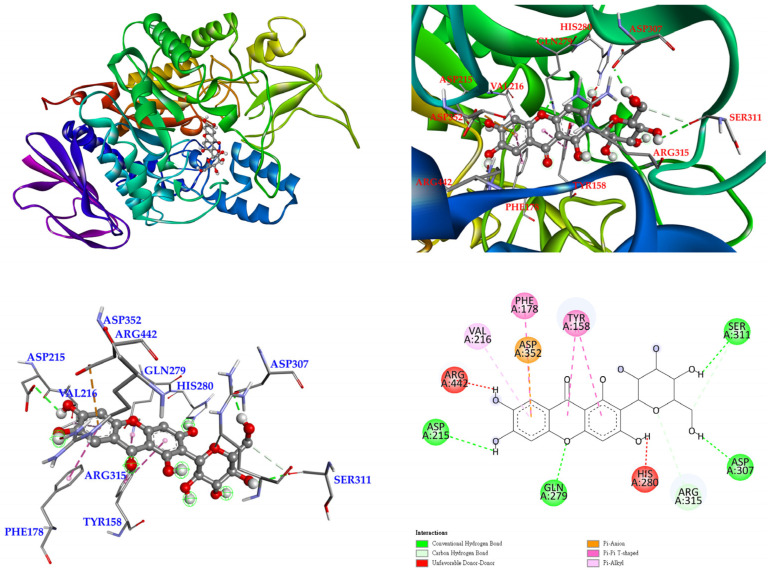
Interaction of mangiferin with active sites of *S. cerevisiae* α-glucosidase.

**Figure 6 antioxidants-11-00385-f006:**
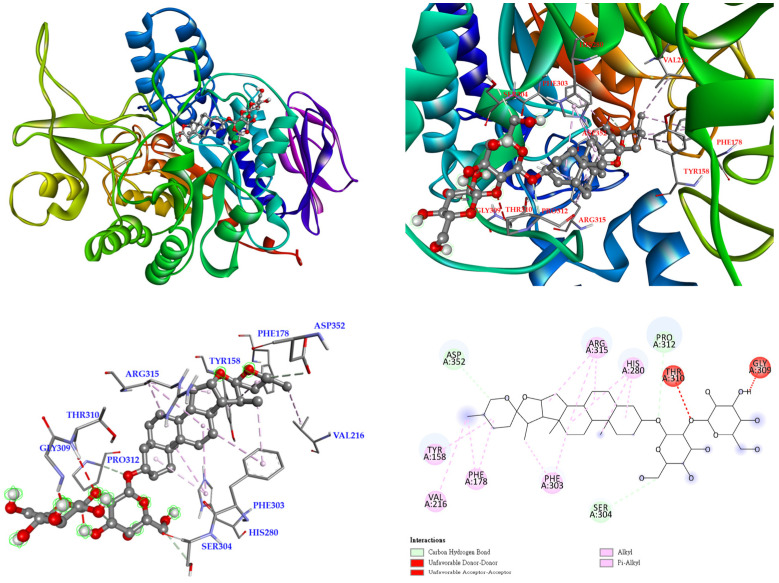
Interaction of timosaponin A-III with active sites of *S. cerevisiae* α-glucosidase.

**Figure 7 antioxidants-11-00385-f007:**
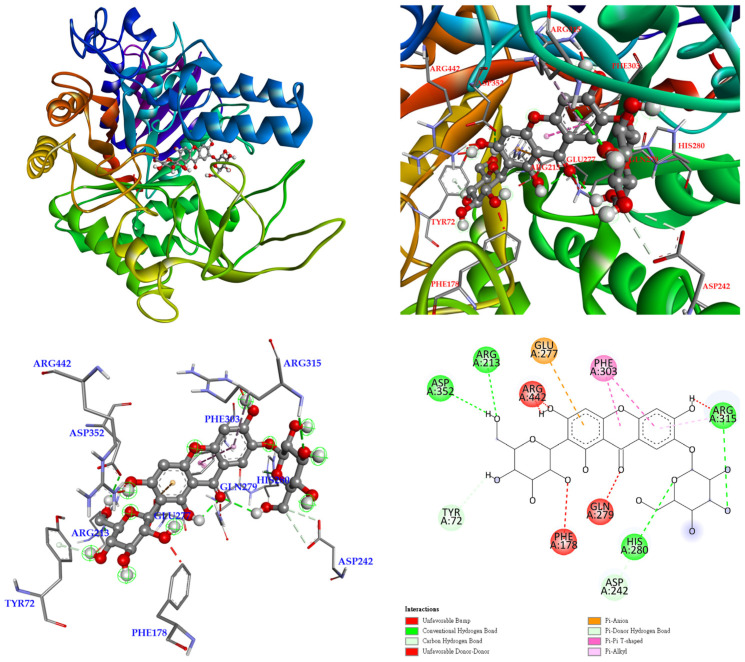
Interaction of isomangiferin with active sites of *S. cerevisiae* α-glucosidase.

**Figure 8 antioxidants-11-00385-f008:**
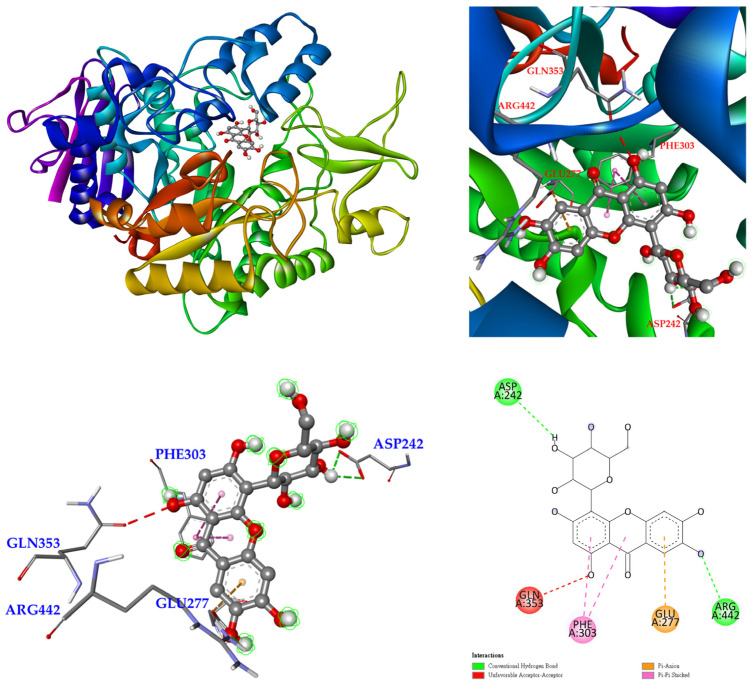
Interaction of neomangiferin with active sites of *S. cerevisiae* α-glucosidase.

**Figure 9 antioxidants-11-00385-f009:**
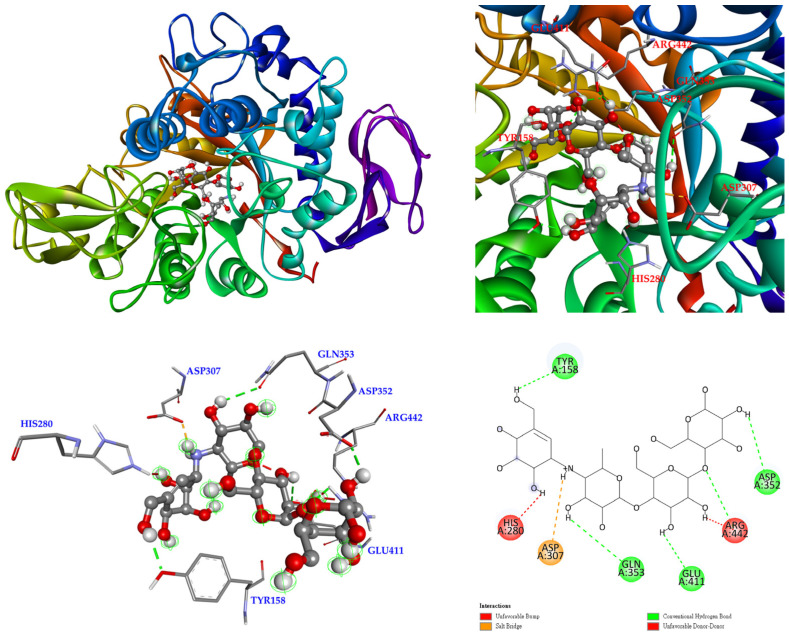
Interaction of acarbose with active sites of *S. cerevisiae* α-glucosidase.

**Table 1 antioxidants-11-00385-t001:** Total phenol, total flavonoid contents, and extraction yields of *Anemarrhena asphodeloides* with each extraction solvent.

Extracting Solvents	TPC (mg/g) ^a^ (GAE)		TFC (mg/g) ^b^ (QCE)		Yields (%) ^c^	
	AA	Salt-Fried AA	AA	Salt-Fried AA	AA	Salt-Fried AA
*n*-Hexane	0	0	42.9 ± 4.9 **	8.3 ± 1.4 *	0.3 ± 0.1	0.3 ± 0.1
Chloroform	23.6 ± 1.9 **	38.6 ± 0.8 **	20.2 ± 2.5 *	24.5 ± 2.7 **	1.6 ± 0.1	0.5 ± 0.1
Dichloromethane	14.9 ± 3.2 *	45.4 ± 2.2 **	24.2 ± 2.7 **	13.3 ± 2.0 *	1.1 ± 0.1	0.3 ± 0.1
Ethyl acetate	21.1 ± 1.8 **	78.2 ± 3.0 ***	5.8 ± 2.2 *	34.9 ± 3.0 **	2.1 ± 0.1	0.6 ± 0.1
Acetone	45.3 ± 1.2 ***	55.2 ± 3.7 **	16.8 ± 2.7 *	45.4 ± 4.2 **	3.0 ± 0.1	1.3 ± 0.1
Ethanol	13.8 ± 1.4 **	33.3 ± 1.1 ***	10.9 ± 2.3 *	23.6 ± 3.8 *	17.7 ± 0.5	18.4 ± 0.7
Methanol	4.8 ± 1.9 *	33.1 ± 2.7 **	8.7 ± 2.2 *	23.8 ± 3.3 *	44.0 ± 1.2	39.2 ± 1.4
Water	18.6 ± 3.7 *	26.3 ± 1.7 **	15.4 ± 2.4 *	23.6 ± 2.6 **	76.7 ± 4.4	72.9 ± 2.5

^a^ Total phenol content (TPC) was expressed in mg of gallic acid equivalents (GAE) per gram of extract. ^b^ Total flavonoid content (TFC) was expressed in mg of quercetin equivalents (QCE) per gram of extract. ^c^ Yield was calculated as % yield = (weight of extract/initial weight of dry sample) × 100. Values are expressed as means ± standard error. * *p* < 0.05; ** *p* < 0.01; *** *p* < 0.001 compared with the control. AA means *Anemarrhena asphodeloides*. Salt-fried AA means salt-fried *Anemarrhena asphodeloides*.

**Table 2 antioxidants-11-00385-t002:** The antioxidant activities of different solvent extracts from *Anemarrhena asphodeloides* determined with DPPH, ABTS, superoxide radical scavenging, and ferric-reducing antioxidant power (FRAP) assays.

Extracting Solvents	DPPH IC_50_ (μg/mL)		ABTS IC_50_ (μg/mL)		Superoxide IC_50_ (μg/mL)		FRAP (mM/g) ^c^ (TE)	
	AA	Salt-Fried AA	AA	Salt-Fried AA	AA	Salt-Fried AA	AA	Salt-Fried AA
*n*-Hexane	>400	>400	305.8 ± 10.4 *	>400	>400	>400	95.9 ± 5.4 **	42.6 ± 5.7 *
Chloroform	>400	>400	73.8 ± 2.3 **	64.3 ± 3.9 **	>400	>400	278.5 ± 7.2 ***	346.3 ± 5.3 ***
Dichloromethane	>400	>400	95.4 ± 5.5 **	40.9 ± 2.5 **	>400	>400	264.5 ± 14.4 **	355.6 ± 5.1 ***
Ethyl acetate	241.2 ± 15.4 *	219.4 ± 6.4 *	75.3 ± 3.7 **	24.1 ± 1.2 **	>400	>400	282.7 ± 7.3 ***	485.5 ± 4.3 ***
Acetone	123.6 ± 3.9 *	169.8 ± 8.6 *	48.5 ± 2.9 **	41.9 ± 0.7 **	>400	>400	381.5 ± 8.3 ***	481.6 ± 19.0 **
Methanol	193.3 ± 9.6 *	181.2 ± 5.2 *	216.0 ± 3.9 *	119.4 ± 4.2 *	>400	>400	152.2 ± 12.8 **	255.2 ± 4.6 ***
Ethanol	150.1 ± 1.1 *	194.2 ± 4.3 *	177.8 ± 3.4 *	112.4 ± 3.6 *	>400	>400	203.2 ± 12.4 **	296.0 ± 11.7 **
Water	127.7 ± 2.3 *	113.9 ± 8.6 *	117.9 ± 0.6 *	97.5 ± 1.6 *	369.9 ± 4.3 *	295.4 ± 6.1 *	230.0 ± 13.8 **	323.7 ± 14.9 **
BHT ^a^	35.5 ± 0.6 **	39.7 ± 1.4 **	20.6 ± 0.2 **	19.6 ± 0.4 **	N.A. ^b^	N.A. ^b^	4297.0 ± 48.4 ***	4216.1 ± 66.5 ***

Results are expressed as half inhibitory concentration (IC_50_) of each free-radical scavenging activity. ^a^ Butylated hydroxytoluene (BHT) used as positive control. ^b^ N.A. indicates not available (poor solubility). ^c^ Ferric-reducing antioxidant power (FRAP) assay was expressed as millimolar (mM) of Trolox equivalents (TE) per gram of extract. * *p* < 0.05; ** *p* < 0.01; *** *p* < 0.001 compared with the control. AA means *Anemarrhena asphodeloides*. Salt-fried AA means salt-fried *Anemarrhena asphodeloides*.

**Table 3 antioxidants-11-00385-t003:** α-Glucosidase inhibitory activities of different solvent extracts.

Extracting Solvents	α-Glucosidase IC_50_ (μg/mL)	
	AA	Salt-Fried AA
*n*-Hexane	25.9 ± 2.0 *	34.8 ± 3.0 *
Chloroform	23.4 ± 2.3 *	34.7 ± 4.7 *
Dichloromethane	21.8 ± 1.4 **	32.0 ± 2.5 *
Ethyl acetate	55.5 ± 2.8 *	26.0 ± 0.6 **
Acetone	101.7 ± 15.9	105.0 ± 2.6 *
Methanol	>200	>200
Ethanol	>200	>200
Water	>200	>200
Acarbose ^a^	319.5 ± 17.3 *	305.0 ± 4.6 *

^a^ Acarbose used as positive control. * *p* < 0.05 and ** *p* < 0.01 compared with the control. AA means *Anemarrhena asphodeloides*. Salt-fried AA means salt-fried *Anemarrhena asphodeloides*.

**Table 4 antioxidants-11-00385-t004:** AChE inhibitory assay of different solvent extracts.

Extracting Solvents	AChE Inhibitory Assay IC_50_ (μg/mL)	
	AA	Salt-Fried AA
*n*-Hexane	163.7 ± 6.7 *	134.2 ± 5.2 *
Chloroform	133.3 ± 6.8	135.6 ± 6.0 *
Dichloromethane	129.5 ± 6.2 *	79.7 ± 2.6 *
Ethyl acetate	169.9 ± 4.0 *	133.8 ± 6.5 *
Acetone	130.2 ± 3.7 *	162.3 ± 6.1 *
Methanol	126.5 ± 4.6 *	126.8 ± 4.6 *
Ethanol	117.1 ± 2.6 *	85.1 ± 6.9 **
Water	139.4 ± 5.2	73.6 ± 2.8 *
Galanthamine ^a^	0.5 ± 0.1 **	0.5 ± 0.1 **

^a^ Galanthamine used as positive control. * *p* < 0.05 and ** *p* < 0.01 compared with the control. AA means *Anemarrhena asphodeloides*. Salt-fried AA means salt-fried *Anemarrhena asphodeloides*.

**Table 5 antioxidants-11-00385-t005:** Identification and quantification of the main active compounds of *A. asphodeloides* in different solvent extracts.

Extracting Solvents	Neomangiferin (mg/kg)	Timosaponin A-III (mg/kg)	Isomangiferin (mg/kg)	Mangiferin (mg/kg)	Total Amount (mg/kg)
Water (AA)	2.5 ± 0.5 *	5.4 ± 0.7 **	2.8 ± 0.4 **	4.4 ± 0.6 **	15.1 ± 0.6 ***
Methanol (AA)	2.0 ± 0.4 *	3.6 ± 0.4 **	3.3 ± 0.4 **	3.8 ± 0.4 **	12.8 ± 1.6 **
Ethanol (AA)	1.6 ± 0.7	3.1 ± 0.4 **	1.0 ± 0.4 *	3.8 ± 0.2 ***	9.6 ± 0.5 ***
Acetone (AA)	7.9 ± 1.1 **	4.8 ± 1.3 *	9.6 ± 1.2 *	6.5 ± 1.9 *	28.8 ± 1.2 ***
Ethyl acetate (AA)	6.2 ± 0.9 **	2.4 ± 0.9 *	2.2 ± 0.8 *	4.2 ± 1.0 *	15.1 ± 0.7 ***
Chloroform (AA)	4.4 ± 1.5 *	13.9 ± 1.2 **	23.7 ± 1.9 **	5.5 ± 1.2 *	47.5 ± 1.5 ***
Dichloromethane (AA)	8.7 ± 1.4 **	4.3 ± 1.4 *	3.3 ± 0.9 *	7.2 ± 1.3 *	23.5 ± 1.3 **
*n*-Hexane (AA)	2.7 ± 1.2	21.3 ± 1.1 ***	12.3 ± 1.9 **	7.6 ± 1.2 **	43.9 ± 1.4 ***
Water (salt-fried AA)	0.9 ± 0.5	3.4 ± 0.8 *	3.8 ± 0.6 **	4.4 ± 1.1 *	12.6 ± 0.8 **
Methanol (salt-fried AA)	3.0 ± 0.7 *	5.6 ± 0.9 **	5.3 ± 0.8 **	7.8 ± 0.7 **	21.8 ± 0.8 ***
Ethanol (salt-fried AA)	1.9 ± 0.9	3.1 ± 0.9 *	3.0 ± 0.8 *	5.8 ± 1.2 *	13.8 ± 1.0 **
Acetone (salt-fried AA)	2.9 ± 1.0 *	3.8 ± 1.2 *	9.6 ± 1.3 **	5.5 ± 1.2 *	21.8 ± 1.2 **
Ethyl acetate (salt-fried AA)	3.2 ± 0.9 *	9.6 ± 1.3 **	4.2 ± 1.6 *	10.2 ± 1.2 **	27.3 ± 1.3 ***
Chloroform (salt-fried AA)	3.4 ± 1.2 *	12.9 ± 1.4 **	23.7 ± 1.7 **	6.5 ± 1.2 *	46.5 ± 1.4 ***
Dichloromethane (salt-fried AA)	2.7 ± 1.6	10.3 ± 1.4 **	3.6 ± 1.2 *	6.2 ± 1.6 *	22.8 ± 1.5 **
*n*-Hexane (salt-fried AA)	12.7 ± 1.2 **	3.3 ± 1.1 *	21.3 ± 1.9 **	4.6 ± 1.2 *	41.9 ± 1.4 ***

Results are expressed as milligrams of each compound in kilogram of extract. * *p* < 0.05; ** *p* < 0.01; *** *p* < 0.001 compared with the blank. AA means *Anemarrhena asphodeloides*. Salt-fried AA means salt-fried *Anemarrhena asphodeloides*.

**Table 6 antioxidants-11-00385-t006:** The antioxidant activities of isolated components from *Anemarrhena asphodeloides* determined with DPPH, ABTS, superoxide radical scavenging, and ferric-reducing antioxidant power (FRAP) assays.

Compounds	DPPH IC_50_ (μg/mL)	ABTS IC_50_ (μg/mL)	Superoxide IC_50_ (μg/mL)	FRAP (mM/g) ^c^ (TE)
Mangiferin	5.4 ± 0.3 *	3.7 ± 0.2 *	53.8 ± 2.4 *	9371.4 ± 183.9 ***
Timosaponin A-III	>200	>200	>200	19.5 ± 3.9 *
Neomangiferin	>200	81.9 ± 4.6 **	>200	83.1 ± 5.3 **
Isomangiferin	16.7 ± 1.1 *	5.6 ± 0.4 *	155.3 ± 13.8 *	6359.9 ± 176.8 ***
BHT ^a^	27.2 ± 1.9 *	15.4 ± 0.9 *	N.A. ^b^	3963.9 ± 104.0 ***

Results are expressed as half inhibitory concentration (IC_50_) of each free-radical scavenging activity. ^a^ Butylated hydroxytoluene (BHT) used as positive control. ^b^ N.A. indicates not available (poor solubility). ^c^ Ferric-reducing antioxidant power (FRAP) assay expressed as millimolar (mM) of Trolox equivalents (TE) per gram of extract. * *p* < 0.05, ** *p* < 0.01, and *** *p* < 0.001 compared with the control.

**Table 7 antioxidants-11-00385-t007:** α-Glucosidase inhibitory activities of pure compounds.

Compounds	α-Glucosidase IC_50_ (μg/mL)
Mangiferin	61.6 ± 5.1 **
Timosaponin A-III	72.5 ± 1.9 *
Neomangiferin	>400
Isomangiferin	183.2 ± 1.3 *
Acarbose ^a^	322.0 ± 18.7 *

^a^ Acarbose used as positive control. * *p* < 0.05 and ** *p* < 0.01 compared with the control.

**Table 8 antioxidants-11-00385-t008:** Acetylcholinesterase (AChE) inhibitory assays of pure compounds.

Compounds	AchE Inhibitory Assay IC_50_ (μg/mL)
Mangiferin	67.8 ± 4.2 **
Timosaponin A-III	132.9 ± 10.9 *
Neomangiferin	70.8 ± 8.5 **
Isomangiferin	96.0 ± 6.1 *
Galanthamine ^a^	0.5 ± 0.1 **

^a^ Galanthamine used as positive control. * *p* < 0.05 and ** *p* < 0.01 compared with the control.

**Table 9 antioxidants-11-00385-t009:** Binding energies of active components and acarbose calculated in silico.

Compounds	Affinity (kcal/mol)
Mangiferin	−10.0
Timosaponin A-III	−8.2
Neomangiferin	−2.0
Isomangiferin	−7.7
Acarbose ^a^	−2.3

^a^ Acarbose used as positive control.

## Data Availability

Data is contained within the article and Appendix A.

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
