# Peer review of "Comparison of Various Solvent Extracts and Major Bioactive Components from Unsalt-Fried and Salt-Fried Rhizomes of *Anemarrhena asphodeloides* for Antioxidant, Anti-α-Glucosidase, and Anti-Acetylcholinesterase Activities"

_antioxidants, 2022, doi:10.3390/antiox11020385_

Round 1

Reviewer 1 Report

The authors revised their work. Most of my comments were taken into account during the preparation of corrected version. However, in my opinion one issue should be addressed and some corrections should be done.

The authors did not add any statistics analysis to Table 5. They justify that in some other papers such analysis was not performed. In my opinion statistically significant differences in values of quantified compounds between samples should be indicated. In many other papers statistical analysis is applied for quantitative data. 

Please show statistically significant differences between values obtained for different extraction methods for quantified compounds.

Author Response

Please see an attached file.

Reviewer 2 Report

N/A

Author Response

Please see an attached file.

Round 2

Reviewer 1 Report

No further comments.

This manuscript is a resubmission of an earlier submission. The following is a list of the peer review reports and author responses from that submission.

Round 1

Reviewer 1 Report

Authors studied  the antioxidant, anti-α-glucosidase, and anti-AChE activitiesof A. asphodeloides and its active extracts and components . The work is conducted using correct experimental methods. Only the way of expressing the results is questionable. It is customary to indicate the error associated with a measurement with a single significant digit, while in all the tables the errors are reported with three or more digits. I recommend correcting these values.As a consequence, the result must also be expressed with a number of digits compatible with the error. For instance:

23.61 ± 1.89             24 ± 2  (correct)

1.07 ± 0.11             1.1 ± 0.1  (correct)

Author Response

Please see an attached file.

Reviewer 2 Report

The paper submitted by Chu and coworkers to Antioxidants is focused on the chemical analysis and evaluation of several bio activities of various extracts from Anemarrhena asphodeloides. In my opinion the paper presents a nice pice of phytochemical work. All experiments seem to be correctly performed and described. All proper positive controls are included in bioassays. Phytochemical analysis is also nicely performed and described. In my opinion the paper should be published in Antioxidants after minor revision.

1) statistical analysis should be added to table 5 - please indicate which samples show statistically significant difference in the content of quantified compounds

2) how the HPLC method used for phytochemical screening was optimized? please explain in the text. Was any validation involved?

3) in my opinion one of the chromatograms should be moved to the main text for the reader's convenience - please consider showing representative chromatogram in the main text

Author Response

Please see an attached file.

Reviewer 3 Report

The manuscript by Chu C-Y et al. reports the antioxidant, anti-α-glucosidase, and anti-AChE activities from the sp Anemarrhena asphodeloides. Also, the authors isolated its solvent extracts and bioactive components.

The abstract, introduction, experimental procedures and results are presented well. However, the present version of the manuscript is incomplete because the authors didn’t include a discussion. Although there is a section of results and discussion but no discussion in the text. A cutting-edge discussion is necessary to include in the manuscript. A discussion is important for any original manuscript to validate the study.

References are not updated. I believe the references will also be updated if the authors can write a good discussion.

I have also a doubt about the originality of this manuscript. I would suggest the authors to clarify this issue.   

Author Response

Please see an attached file.

Reviewer 4 Report

The manuscript by Chen et al. concerns the investigation the biological properties (including antioxidant, anti-acetylcholinesterase, and anti-α-Glucosidase activities) of extracts from the Rhizome of Anemarrhena asphodeloides, an Asian plant. At the same time, each of this extract represents a mixture of compounds. It should be noted, that Authors didn’t explain any regularity of extractions (for example, relationships between solvent polarity vs. polarity of extracted compounds; surprising differences in behavior of related CHCl3 and CH2Cl2; surprising high extracting properties of n-hexane). The results obtained look as routine ones; the antioxidant activities of these “cocktails” are low. As Authors stated, this plant is used in traditional Chinese medicine; therefore, it is evident, that it has a biological action. It should be noted, that in modern medicine, the properties of individual compounds should be studied and, therefore, applied. From these points of view, the scientific significance of the manuscript is weak.

Another weak point of the manuscript consists in presence of two weakly connected parts, i.e. investigation of the extracts and individual components, including their docking. This docking concerns the connectivity of individual compounds (mangiferin, timosaponin A-III, isomangiferin, neomangiferin; interestingly, it is known, that each of this compound has antioxidant properties, see, for example LWT - Food Sci. . Technol., 2013, 51(1), 129; https://www.sciencedirect.com/topics/pharmacology-toxicology-and-pharmaceutical-science/isomangiferin; J. Pharm Biomed. Anal., 2020, 191, 113616) with S. cerevisiae α-glucosidase. The second part represents a subject for the investigation.

As a conclusion, in present view this manuscript cannot be accepted for publication. I recommend Rejection.

Author Response

Please see an attached file.

Round 2

Reviewer 3 Report

The authors revised some and clarified the originality of their fundings. However, I am still not convinced with their discussion section. There are no discussions included yet in some sections. The authors require to include a discussion section with updated relevant references. I would suggest separating the discussion section from the results section. 

The authors still have a lot of works for setting a nice discussion section and for completing it a major revision is required.  

Reviewer 4 Report

The manuscript of Chen et al. after revision was improved in some extent. Several literature sources were inserted by the Authors. But I think that my questions aroused previously were answered only partly. I didn’t understand why comparative analysis of activity of mixtures represents an academic interest. I didn’t understand how the extraction data presented in the manuscript might affect further research.

For my opinion, as I stated previously, the one of the advantage of this manuscript is an investigation of antioxidant activity of individual components of extracts. This and the presented composition of this extracts may be subjects for publication. It should be noted that antioxidant activity of several components is known, what requires more detailed literature analysis.

I think that in present view manuscript should be rejected.